# Synchronous Breast and Cervical Carcinoma: A Genetic Point of View

**DOI:** 10.3390/biomedicines11020525

**Published:** 2023-02-11

**Authors:** Maya Mazuwin Yahya, Mohd Pazudin Ismail, Shogeta Ramanathan, Muhammad Nashriq Kadir, Azzahra Azhar, Noorul Balqis Che Ibrahim, Chee Lee Wee, Zahiah Mohd Amin, Seng Kong Tham, Shuhaila Mat-Sharani, Nik Soriani Yaacob

**Affiliations:** 1Department of Surgery, School of Medical Sciences, Universiti Sains Malaysia, Health Campus, Kubang Kerian 16150, Kelantan, Malaysia; 2Breast Cancer Research and Awareness Unit (BestARi), Hospital Universiti Sains Malaysia, Kubang Kerian 16150, Kelantan, Malaysia; 3Department of Obstetrics and Gynaecology, School of Medical Sciences, Universiti Sains Malaysia, Health Campus, Kubang Kerian 16150, Kelantan, Malaysia; 4Department of Pathology, School of Medical Sciences, Universiti Sains Malaysia, Health Campus, Kubang Kerian 16150, Kelantan, Malaysia; 5Department of Chemical Pathology, School of Medical Sciences, Universiti Sains Malaysia, Health Campus, Kubang Kerian 16150, Kelantan, Malaysia; 6MyGenome Sdn Berhad, Kuala Lumpur 50400, Selangor, Malaysia; 7Biomedicine Program, School of Health Sciences, Universiti Sains Malaysia, Health Campus, Kubang Kerian 16150, Kelantan, Malaysia

**Keywords:** metastatic breast cancer, cervical cancer, concurrent chemoradiotherapy, gene expression profiling

## Abstract

Breast carcinoma is the most common cancer of women in Malaysia. The most common sites of metastasis are the lung, liver, bone and brain. A 45-year-old lady was diagnosed with left invasive breast carcinoma stage IV (T4cN1M1) with axillary lymph nodes and lung metastasis. She was noted to have a cervical mass through imaging, and biopsy showed CIN III. Post chemotherapy, the patient underwent left simple mastectomy with examination under anaesthesia of the cervix, cystoscopy and staging. The cervical histopathological examination (HPE) showed squamous cell carcinoma, and clinical staging was 2A. The breast tissue HPE showed invasive carcinoma with triple receptors positivity. The patient was given tamoxifen and put on concurrent chemoradiotherapy (CCRT) for the cervical cancer. The management of each pathology of this patient involved a multi-disciplinary team that included surgeons, oncologists, gynaecologists, pathologists and radiologists. Due to the complexity of the case with two concurrent cancers, the gene expression profiles may help predict the patient’s clinical outcome.

## 1. Introduction

Breast cancer is the most common cancer in Malaysia, while cervical cancer is the fourth most common cancer among women in Malaysia [1]. The common sites for invasive breast cancer to metastasize are the lung, liver, bone and brain [2] while for cervical cancer, the frequent metastatic sites are the lungs, extra-pelvic nodes, liver and bones [3]. The most common histological finding for cervical cancer is squamous cell carcinoma. Both breast and cervical cancers are detected late, and many cancer patients do not receive care consistent with global standards. The current literature shows rare occurrence of concurrent breast cancer with cervical cancer, probably due to the contrasting predisposing factors [4]. We present a patient with dual pathology for breast with cervical cancer and the gene expression profiles obtained following palliative chemotherapy and concurrent chemotherapy and radiation (CCRT).

## 2. Case Presentation

Despite breast cancer being a common cancer in Malaysia, a concurrent presentation with cervical cancer is rare. Here, we present a case with a concurrent breast and cervical cancer with additional information on gene expression profiles following palliative chemotherapy and CCRT. A 45-year-old Malay lady with a history of two miscarriages and a life birth presented with firm, non-mobile left breast lump of 8 × 4 cm with no nipple discharge. She was diagnosed with invasive carcinoma with estrogen receptor (ER), progesterone receptor (PR) and human epidermal growth factor receptor 2 (HER2) positive. She had no known risk factors for breast cancer such as strong family history, early menarche nor late age at first pregnancy.

The staging of the disease showed axillary lymphadenopathy (up to level III) and diffused lung nodules suggestive of metastasis (T4cN1M1-stage IV). Uterus was bulky with hematometra. Per speculum vaginal examination showed cervix with confined fleshy growth at 9 to 3 o’clock position measuring 4 × 3 cm, with contact bleeding. Punch biopsy showed cervical intraepithelial neoplasia (CIN) stage III; however, invasion could not be ruled out. The patient was started on chemotherapy including 5-fluorouracil, epirubicin and cyclophosphamide, for six cycles for the breast cancer.

Reassessment computed tomography (CT) scan post chemotherapy for the breast cancer showed reduced breast lesion from 2.9 × 4.0 × 5.7 cm to 1.5 × 2.8 cm with no effect on the cervical mass. The patient underwent left simple mastectomy, and histopathological examination showed the same receptor status and morphological features as per biopsy (Figure 1). There were still abundant viable malignant epithelial cells with arranged tubules, cords and trabeculae pattern, exhibiting only a minor loss of neoplastic cells post neo-adjuvant chemotherapy. The final diagnosis was invasive breast carcinoma of no special type with grade 1 Modified Bloom Richardson grading system.

An examination under anaesthesia noted normal vulva and smooth vagina with no nodularity. However, a small raw area (1 × 1 cm) was noted at the upper 1/3 of the vagina (Figure 2A). An exophytic growth of 4 × 4 cm, occupied 9 to 5 o’clock position of the friable cervix which bled on touch (Figure 2B). Cystoscopy noted normal bladder mucosa and no evidence of tumour invasion. The assessment showed clinical staging of the cervical cancer as stage 2A. Histopathological examination of the upper 1/3 posterior vagina and cervix was consistent with human papillomavirus (HPV)-associated keratinizing squamous cell carcinoma (Figure 3). The tumour cells showed block positivity towards P16 immunohistochemical stains (not shown).

The patient was started on tamoxifen therapy for the breast cancer. A CT scan at 2 months post-op showed no local recurrence of the left breast carcinoma nor contralateral involvement, and showed resolution of the lung metastasis (Figure 4). However, the cervical carcinoma size increased with adjacent inflammatory changes, consistent with progressive disease (Figure 5A). No pelvic lymphadenopathy and no distant metastases in the abdomen were noted. The patient then completed CCRT consisting of 25 fractions of external beam radiation and weekly cisplatin 40 mg/m^2^, with additional 8 fractions of external beam radiation as booster. The size of the cervix was consequently reduced (Figure 5B); no tumour mass was noted on the cervix and vagina. In addition, no malignant cells were observed in the cervical smear.

## 3. Gene Expression Profiling

Blood samples were obtained from the patient two months post mastectomy, after completion of the palliative therapy (Sample 1) and following completion of the CCRT (Sample 2). At this stage, the patient was considered cancer-free. Blood samples were collected in Tempus^TM^ Blood RNA Tubes (Thermo Fisher Scientific, Waltham, MA, USA), and the total RNA was extracted using a Tempus^TM^ Spin RNA Isolation Kit (Thermo Fisher Scientific, Waltham, MA, USA). Gene expression analysis was then performed using the nCounter^®^ SPRINT Profiler (NanoString Technologies, Seattle, WA, USA), in triplicates. A total of 625 genes were profiled for Samples 1 and 2, against seven normal control samples. The control subjects were female of more than 40 years old, with no family cancer history or other critical or chronic diseases and were not on any medication.

Data for all samples were processed and normalized using nSolver analysis software (version 4.0, NanoString Technologies, USA) followed by differential gene expression analysis using advanced nSolver analysis. Differentially expressed genes (DEGs) with a two-fold change or more are summarized in Table 1 and Table 2. Gene annotations were obtained from NCBI gene ID “https://www.ncbi.nlm.nih.gov/gene/ (assessed on 27 March 2022)”. *CEBPE* (CCAAT/enhancer binding protein, epsilon)*, CD24* and *CLC* (Charcot-Leyden crystal) genes were highly upregulated in Sample 2 with 5-, 5.9- and 6.8-fold change, respectively. Results also show that both Sample 1 and Sample 2 display different gene profiles with only seven genes that are commonly downregulated in both (Figure 6). Two of these genes, *CHI3L1* (Chitinase-3 like-protein-1) and *DEFA1* (Defensin Alpha 1), were further downregulated in Sample 2 compared to Sample 1 (from 6.6- to 8.9-fold and 2.2- to 8.5-fold, respectively).

## 4. Discussion

Based on the Malaysian National Cancer registry report, the incidence of female breast cancer has been increasing from 18,206 in 2007–2011 to 21,634 in 2012–2016, accounting for 34.1% of all cancers among females [5]. Delay in seeking medical examination of breast symptoms is a significant problem associated with a lower breast cancer survival rate [6]. The delay in presentation is multifactorial, but some documented reasons include the use of alternative therapy, breast ulcer, palpable axillary lymph nodes, false-negative diagnostic test, non-cancer interpretation and negative attitude towards treatment [7]. Diagnosis is conducted with triple assessment, namely clinical, imaging and tissue biopsy. Surgical management of invasive breast carcinoma with axillary lymph node involvement requires breast conservation or mastectomy and axillary clearance. National comprehensive cancer network (NCCN) guidelines recommend neoadjuvant systemic therapy in women with inoperable breast cancer. It can render inoperable cancer to resectable cancer [8]. The role of chemotherapy, radiotherapy or hormonal therapy depends on the size of the primary tumour, the receptor status and nodal involvement [8]. According to the St. Gallen Consensus 2011, molecular subtypes of breast cancer can be classified into Luminal A (ER+/PR+/HER2-/lowKi-67); Luminal B (ER+/PR+/HER2-/+/high Ki-67); HER2-overexpression (ER-/PR-/HER2+); and triple negative breast cancers/TNBCs (ER-/PR-/HER2-). The basal-like subtype of breast cancer referred to as TNBC was found to be positive for basal marker (CK5/6) expression [9]. The prognosis is poorer in triple negative and HER2-enriched disease. The paradigm had shifted in these two subtypes in which the early-stage disease patients are subjected to neoadjuvant systemic therapy prior to surgery [10]. Hormonal therapy with the selective ER modulator, tamoxifen, is indicated for hormone receptor-positive patients and can be used in both premenopausal and post-menopausal women. On the other hand, selective aromatase inhibitors such as anastrozole and letrozole can be used as hormonal therapy for post-menopausal women [11]. Patients positive for ER, PR and HER2 are usually treated with neoadjuvant chemotherapy with adjuvant endocrine therapy, as well as an additional targeted therapy with trastuzumab for the HER2 positivity.

The case reported herein is an invasive breast carcinoma with luminal B triple positivity subtype, which is a common pathology in breast carcinoma. However dual pathology in a patient with both breast and cervical carcinoma is rare [4]. The patient had metastatic disease with the presence of lung nodules. She was therefore subjected to palliative chemotherapy to reduce the size of the tumour, axillary lymph node and lung nodules, after which she underwent mastectomy with axillary clearance. Patients with ER- and PR-positive breast carcinoma have improved prognosis with palliative chemotherapy, and evidence indicates that hormonal therapy for these patients also improves their quality of life [12].

There is currently no clear association between breast and cervical cancer. For cervical cancer, the management is based on the cancer stage, grade and histopathological type as well as general status of the patient. The treatment may include surgical resection, radiotherapy, chemotherapy or their combination. Our patient was also diagnosed with stage 2A cervical cancer and was counselled for radical hysterectomy. However, the patient refused surgical intervention in view of on-going chemotherapy for her breast cancer. Hence, the patient was given concurrent chemoradiation as the primary treatment for her cervical cancer. Primary chemoradiation is now often used to treat locally advanced cervical cancer, similar to that seen in this patient [13]. The treatment is based on combination therapy with platinum-based regimens and radiation that involves external beam and high-dose intracavity brachytherapy. Post-treatment hysterectomy is not associated with increased survival rates, and, therefore, it is generally not recommended [14]. However, hysterectomy may be performed in patients who have large, bulky tumours or high post-treatment tumour volumes [15].

Advances in genomics have paved the way for further understanding of cancer progression, and the identification of specific genomic patterns could help in diagnosis, prognosis and prediction of treatment response. Cluster of differentiation (CD) markers have a role in cancer diagnosis, choice of treatment and therapeutic monitoring. For example, CD24 is highly expressed in various cancers but is rarely expressed in normal tissues [16,17]. CD24 is a small mucin-like antigen on the cell surface with highly variable glycosylation [18]. The expression of CD24 was reported as an independent, unfavourable prognostic factor for disease-free survival of breast cancer patients [16], especially for those with luminal A and triple-negative subtypes [19]. Its high expression is also associated with aggressive breast cancer and poor survival of early-stage breast cancer patients [19]. Herein, *CD24* is also upregulated in our patient with luminal B triple positivity subtype, following CCRT, and this could be associated with poor prognosis. Interestingly, CD24 transcription is regulated by ERalpha in breast cancer cells [20]. The expression of *CD177* is reduced in our patient, which could also predict poor outcome. *CD177* expression has been reported to be associated with a better prognosis in breast and other cancers [21]. CD177 is expressed in tumour-infiltrating, regulatory T (Treg) cells in breast, renal, lung and colorectal cancers. Treg cells suppress a variety of immune cells to maintain homeostasis and peripheral tolerance, and attenuation of this suppressive activity is associated with an increased anti-tumour immune response [22].

On the other hand, the expression of the transcription factor for granulocyte differentiation, *CEBPE*, was highly upregulated in our patient upon completion of CCRT. Datasets analysis indicates that CEBPE expression predicts better survival rate for patients with acute myeloid leukaemia [23]. CLCs are normally associated with eosinophilic diseases and have a role in type 2 immunity [24]. An upregulation of the *CLC* gene in the current patient may thus indicate an increased tissue-level defence mechanism against the cancer.

The expression of *CHI3L1* and *DEFA1* was downregulated after palliative chemotherapy and then further reduced following CCRT. CHI3L1 signalling promotes tumour progression by promoting cancer cell growth, proliferation, migration, invasion, metastasis, angiogenesis, activation of tumour-associated macrophages and Th2 polarisation of CD4+ T cells [25,26]. Increased CHI3L1 levels in breast and gastric cancer patients promote lung metastasis via activation of the mitogen-activated protein kinase (MAPK) signalling and upregulation of matrix metalloproteinase (MMP) genes [27]. Rusak et al. [28] further reported a positive correlation between CHI3L1 expression and markers of angiogenesis in invasive ductal breast carcinoma, with higher expression observed in triple-negative cases. Hence, considering the significant role of CHI3L in cancer progression and metastasis, the low *CHI3L1* expression in the current patient reflects an effective treatment. The expression of *DEFA1* was also further reduced following CCRT. DEFA1 belongs to the α-defensin family that possesses antimicrobial and immunomodulatory functions, as well as antitumour activities [29]. *DEFA1* and *DEFA3* encode human neutrophil peptides, HNP1-3 [30], which contribute to tumour progression and invasion and, hence, are often detected in elevated levels in cancer patients [29]. The expression of *DEFA1* and *DEFA3* RNA transcripts in blood has been associated with improved response to docetaxel therapy in castration-resistant prostate cancer, suggesting their predictive value for the selection of personalized therapy [31].

The mainstay management of invasive breast carcinoma with axillary lymph node is surgical intervention through removal of the primary tumour with clear margins and axillary surgery. Locally advanced and metastatic diseases are given systemic neoadjuvant chemotherapy to reduce the size of the primary tumour and metastatic nodules for the surgical intervention decision. Responsive primary tumours have shown survival benefits.

In cervical cancer, therapeutic approaches must be carefully tailored to obtain the best outcome for each patient. The treatment options for early stage of the disease involve surgery, radiotherapy and chemotherapy. However, in advanced stage (Stage 2B and above), the treatment is limited to radiotherapy and chemotherapy.

## 5. Conclusions

Patients with both conditions need to be addressed carefully. The prognosis of each individual disease should be considered prior to a decision for definitive treatment. The above patient had stage IV breast cancer and stage 2A cervical cancer. She had undergone aggressive treatment for both conditions, and the one-year post-treatment showed no progression of disease. The genetic findings are by no means conclusive, but the expression profile of a combination of different genetic markers may be useful in the selection of personalized therapy as well as in predicting long-term survival outcome.

## Figures and Tables

**Figure 1 biomedicines-11-00525-f001:**
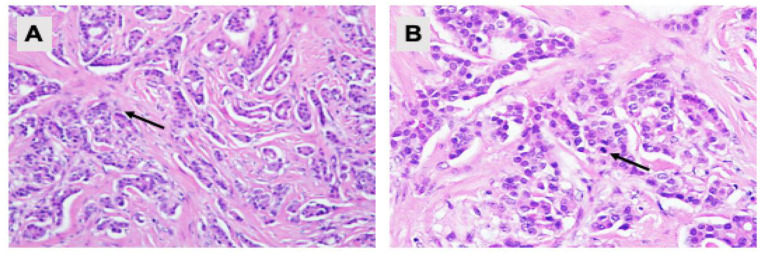
Pathological examination of the left breast tissue. Hematoxylin and eosin staining showing (**A**) malignant tumour cells arranged in tubules and cords (200× magnification); and (**B**) neoplastic cells with mild nuclear pleomorphism, round to oval nuclei, vesicular to hyperchromatic chromatin pattern and eosinophilic cytoplasm. Mitotic activities are rarely seen (arrow) (400× magnification).

**Figure 2 biomedicines-11-00525-f002:**
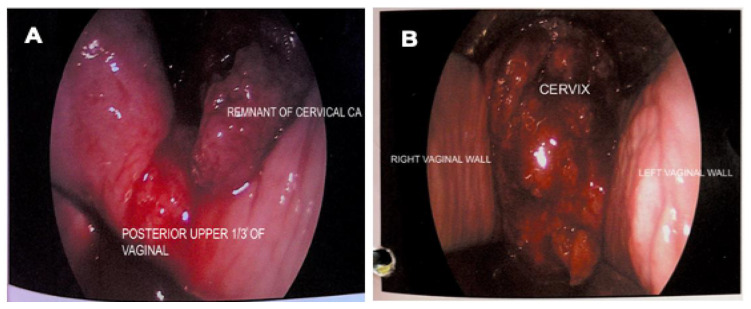
Vaginal and cervical examination. (**A**) Remnant cervical growth and small raw area (1 × 1 cm) at posterior upper 1/3 of vagina; (**B**) Exophytic cervical cancer (4 × 4 cm) occupying 9 to 5 o’clock position of the cervix, was friable, and bled on touch.

**Figure 3 biomedicines-11-00525-f003:**
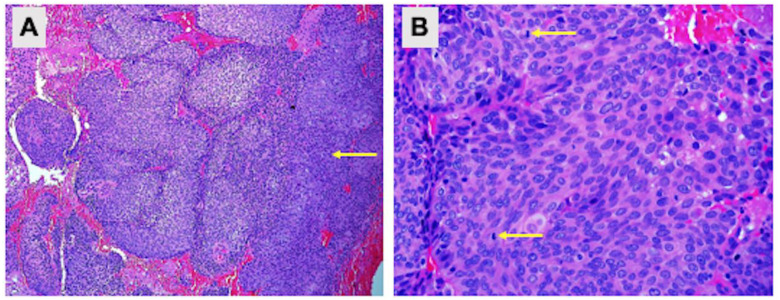
Pathological examination of the upper 1/3 posterior vagina and cervix. Hematoxylin and eosin staining showing (**A**) cell carcinoma infiltration (100× magnification) and (**B**) cells with enlarged hyperchromatic nuclei, inconspicuous nucleoli and moderate eosinophilic cytoplasm. Mitotic activities are brisk (arrow) (400× magnification).

**Figure 4 biomedicines-11-00525-f004:**
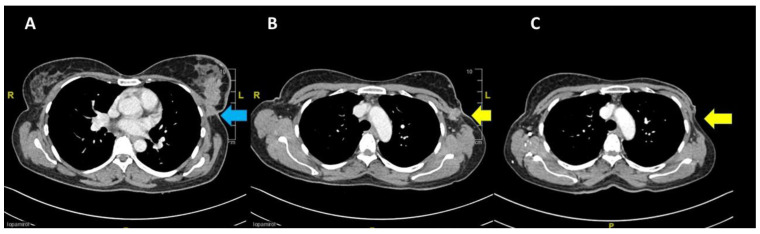
Pre-chemotherapy CT scan showing (**A**) the primary breast lesion (blue arrow) and (**B**) the level of arch of the aorta (yellow arrow indicates fixed nodes to the pectoral muscle laterally); (**C**) post mastectomy at 2 months showing no local recurrence at the left breast.

**Figure 5 biomedicines-11-00525-f005:**
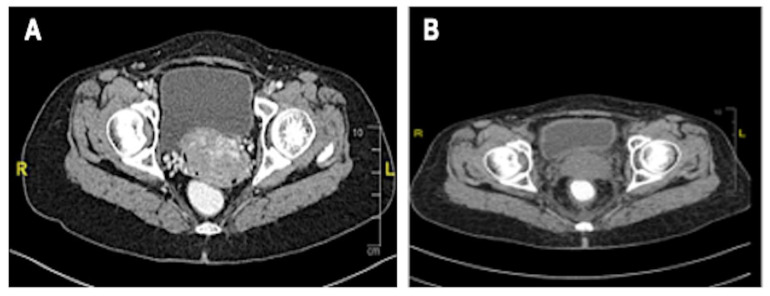
CT scan of the cervix. (**A**) Increased size of cervical mass post chemotherapy for breast cancer, no clear plane with the urinary bladder; (**B**) Reduced size of cervix and clear bladder plane post-CCRT.

**Figure 6 biomedicines-11-00525-f006:**
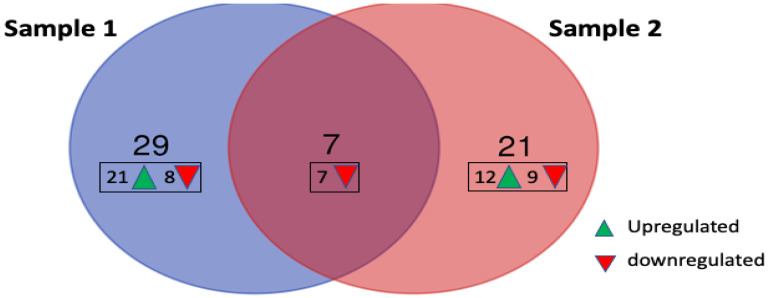
Number of upregulated and downregulated genes in patient samples 1 and 2 compared to the control group. Seven of the genes were downregulated in both samples, but none of the upregulated genes were similarly observed in both.

**Table 1 biomedicines-11-00525-t001:** DEGs after completion of palliative chemotherapy and mastectomy (Sample 1).

No.	Gene	Annotation
Downregulated Genes		
1	* CHI3L1	chitinase-3 like-protein-1
2	* EPB42	erythrocyte membrane protein band 4.2, transcript variant 1
3	MS4A3	membrane spanning 4-domains A3
4	GYPB	glycophorin B (MNS blood group)
5	* PF4	platelet factor 4
6	ALAS2	5′-aminolevulinate synthase 2
7	FOXO4	Homo sapiens forkhead box O4
8	CA1	carbonic anhydrase I
9	* COL18A1	Homo sapiens collagen type XVIII alpha 1 chain
10	SNCA	Homo sapiens synuclein alpha, transcript variant 3
11	* DEFA1	Homo sapiens defensin, alpha 1
12	MYLK	myosin light chain kinase
13	* CAMP	cathelicidin antimicrobial peptide
14	* F2RL1	Homo sapiens coagulation factor II (thrombin) receptor-like 1
15	FKBP1B	FK506 binding protein 1B
Upregulated Genes		
1	CST3	Homo sapiens cystatin C
2	CD3G	CD3g molecule
3	PARP	Homo sapiens poly(ADP-ribose) polymerase 1
4	CENPF	centromere protein F
5	EZH2	enhancer of zeste 2 polycomb repressive complex 2 subunit
6	ENO1	Homo sapiens enolase 1, (alpha)
7	FCN1	Homo sapiens ficolin 1
8	PCNA	proliferating cell nuclear antigen
9	CDK6	cyclin dependent kinase 6
10	HP	Haptoglobin
11	CD5	CD5 molecule
12	PDCD5	Homo sapiens programmed cell death 5
13	PEBP1	Homo sapiens phosphatidylethanolamine binding protein 1
14	FEN1	flap structure-specific endonuclease 1
15	MIF	macrophage migration inhibitory factor
16	LYZ	Homo sapiens lysozyme
17	CXCR3	C-X-C motif chemokine receptor 3
18	SQLE	squalene epoxidase
19	MIER2	MIER family member 2
20	ASGR2	asialoglycoprotein receptor 2
21	VCAN	Homo sapiens versican, transcript variant 1

**Note**: * marks the genes that are significantly reduced in both samples 1 and 2.

**Table 2 biomedicines-11-00525-t002:** DEGs following completion of CCRT, and patient is tumour-free (Sample 2).

No.	Gene	Annotation
Downregulated Genes		
1	* CHI3L1	chitinase-3 like-protein-1
2	* DEFA1	Homo sapiens defensin, alpha 1
3	* PF4	platelet factor 4
4	ANXA3	annexin A3
5	MMP9	matrix metallopeptidase 9
6	CD177	CD177 molecule
7	* CAMP	cathelicidin antimicrobial peptide
8	* F2RL1	Homo sapiens coagulation factor II (thrombin) receptor-like 1
9	EPB42	erythrocyte membrane protein band 4.2, transcript variant 1
10	* DEFA4	Homo sapiens defensin, alpha 4, corticostatin
11	LRG1	leucine rich alpha-2-glycoprotein 1
12	* COL18A1	collagen type XVIII alpha 1 chain
13	IL1R2	Homo sapiens interleukin 1 receptor, type II, transcript variant 2
14	TREM1	Homo sapiens triggering receptor expressed on myeloid cells 1, transcript variant 2
15	NAMPT	nicotinamide phosphoribosyltransferase
16	S100A8	S100 calcium binding protein A8
Upregulated Genes		
1	ISG15	ISG15 ubiquitin-like modifier
2	COL9A2	collagen type IX alpha 2 chain
3	SOCS1	Homo sapiens suppressor of cytokine signaling 1
4	IL2R	Homo sapiens interleukin 2 receptor
5	CSF1	colony stimulating factor 1 (macrophage)
6	C3AR1	complement component 3a receptor 1
7	FGFR2	fibroblast growth factor receptor 2
8	P2RY14	Homo sapiens purinergic receptor P2Y, G-protein coupled, 14
9	EEF2K	Homo sapiens eukaryotic elongation factor 2 kinase
10	CEBPE	CCAAT/enhancer binding protein, epsilon
11	CD24	Homo sapiens CD24 molecule
12	CLC	Homo sapiens Charcot-Leyden crystal protein

**Note**: * marks the genes that are significantly reduced in both samples 1 and 2.

## Data Availability

The data sets generated and/or analysed during the current study are available from the corresponding author on reasonable request.

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
