# Peer review of "Synchronous Breast and Cervical Carcinoma: A Genetic Point of View"

_biomedicines, 2023, doi:10.3390/biomedicines11020525_

Round 1

Reviewer 1 Report

As for a case report, this manuscript is quite complete.

Line 36, not “commonest” but “most common”

Line 36, are you sure this is true (breast cancer in men)? Also, the proper reference should be added.

Line 37, remove “most”

Line 46, so what is the aim of such presentation? A one sentence here is needed.

Line 51, what were those factors (that were excluded, in this case)

Line 144, a reference is needed here

Line 167, what about this work: “ 10.1371/journal.pone.0240788 “ (this is not mine article nor any of my colleagues).

Author Response

Thank you for the reviewers’ comments and suggestions for improvement. The changes had been made accordingly. Below are the authors’ responses to the reviewers’ comments.

Reviewer 1:

  1. Line 36 – changed to ‘most common’
  2. Line 36 – removed ‘among both genders’
  3. Line 37 – removed ‘most’
  4. Line 46 (now line 48 – 51) - an introduction to the case is added ‘Despite breast cancer being a common cancer in Malaysia, a concurrent presentation with cervical cancer is rare. Here, we present a case with a concurrent breast and cervical cancer with additional information on the gene expression profiles following palliative chemotherapy and CCRT.’
  5. Line 51 (now line 55) – the sentence has been modified to ‘She had no risk factors for breast cancer such as strong family history, early menarche nor late age at first pregnancy.’
  6. Line 144 (now line 150) – reference no. 8 is cited.
  7. Line 167 – The authors consider that the suggested article is not relevant to the discussion of the current submission. The article reports on awareness of risk and symptoms of both breast and cervical cancers among the selected population in South Africa and Uganda. No information related to cases on concurrent breast and cervical cancer was included.

Reviewer 2 Report

1. In tables 1 and 2, the number of genes does not match with figure 6, please check.

2. Based on the data in Figure 6, 7 genes are equally reduced in both samples, it would be appropriate to list them separately.

3. You can add a column in tables 1 and 2 indicating for which cancer this gene is significant, if such information is available.

Author Response

Thank you for the reviewers’ comments and suggestions for improvement. The changes had been made accordingly. Below are the authors’ responses to the reviewers’ comments.

Reviewer 2:

  1. The no. of genes stated in Figure 6 is correct and tallies with the no. of genes listed in both Tables 1 and 2. Figure 6 shows that there are 7 overlapping genes between both samples, meaning that the total no. of DEGs for each sample would include these overlapping genes. Hence, the total DEGs is 36 for Sample 1 and 28 for Sample 2.
  2. Thank you for the suggestion. The 7 genes that are equally reduced are marked with ‘*’ in both Tables 1 and 2. And a note is to be added under each table as below. We hope this is acceptable instead of creating a separate new table

Note: * marks the genes that are significantly reduced in both samples 1 and 2

  1. Thank you for the suggestion. In our study case, blood sample 1 was obtained after the patient had already completed mastectomy and chemotherapy for breast cancer and for sample 2, the patient was already considered tumor-free. We do not have the patient sample prior to receiving any treatments for comparison, to reflect genes that are potentially significant in breast or cervical cancer development in this case. Nevertheless, we have included in the discussion a few DEGs observed, that have been previously reported in the literature to have significant roles in cancer progression, diagnosis, management, or prognosis.

The authors thank the reviewers for their valuable comments, and we hope that the changes made are satisfactory. Please update us if any other additional corrections are needed.  Thank you for considering our article in your esteemed journal.

Round 2

Reviewer 2 Report

I have no comments on the article. I believe that in its present form the manuscript can be recommended for publication.